# Identification of the HNSC88 Molecular Signature for Predicting Subtypes of Head and Neck Cancer

**DOI:** 10.3390/ijms241713068

**Published:** 2023-08-22

**Authors:** Yi-Hsuan Chuang, Chun-Yu Lin, Jih-Chin Lee, Chia-Hwa Lee, Chia-Lin Liu, Sing-Han Huang, Jung-Yu Lee, Wen-Sen Lai, Jinn-Moon Yang

**Affiliations:** 1Institute of Bioinformatics and Systems Biology, National Yang Ming Chiao Tung University, Hsinchu 300, Taiwan; 2Center for Intelligent Drug Systems and Smart Bio-Devices, National Yang Ming Chiao Tung University, Hsinchu 300, Taiwan; 3Department of Otolaryngology—Head and Neck Surgery, Tri-Service General Hospital, National Defense Medical Center, Taipei 114, Taiwan; 4School of Medical Laboratory Science and Biotechnology, College of Medical Science and Technology, Taipei Medical University, Taipei 110, Taiwan; 5TMU Research Center of Cancer Translational Medicine, Taipei Medical University, Taipei 110, Taiwan; 6Ph.D. Program in Medicine Biotechnology, College of Medical Science and Technology, Taipei Medical University, Taipei 110, Taiwan; 7Graduate Institute of Life Sciences, National Defense Medical Center, Taipei 114, Taiwan; 8Department of Otolaryngology—Head and Neck Surgery, Taichung Armed Forces General Hospital, Taichung 404, Taiwan; 9Center of Excellence for Metabolic Associated Fatty Liver Disease, National Sun Yat-Sen University, Kaohsiung 804, Taiwan

**Keywords:** head and neck cancer, systems biology, molecular subtypes, molecular signature, human papillomavirus, prognostic biomarker

## Abstract

Head and neck squamous cell carcinoma (HNSC) exhibits genetic heterogeneity in etiologies, tumor sites, and biological processes, which significantly impact therapeutic strategies and prognosis. While the influence of human papillomavirus on clinical outcomes is established, the molecular subtypes determining additional treatment options for HNSC remain unclear and inconsistent. This study aims to identify distinct HNSC molecular subtypes to enhance diagnosis and prognosis accuracy. In this study, we collected three HNSC microarrays (*n* = 306) from the Gene Expression Omnibus (GEO), and HNSC RNA-Seq data (*n* = 566) from The Cancer Genome Atlas (TCGA) to identify differentially expressed genes (DEGs) and validate our results. Two scoring methods, representative score (RS) and perturbative score (PS), were developed for DEGs to summarize their possible activation functions and influence in tumorigenesis. Based on the RS and PS scoring, we selected candidate genes to cluster TCGA samples for the identification of molecular subtypes in HNSC. We have identified 289 up-regulated DEGs and selected 88 genes (called HNSC88) using the RS and PS scoring methods. Based on HNSC88 and TCGA samples, we determined three HNSC subtypes, including one HPV-associated subtype, and two HPV-negative subtypes. One of the HPV-negative subtypes showed a relationship to smoking behavior, while the other exhibited high expression in tumor immune response. The Kaplan–Meier method was used to compare overall survival among the three subtypes. The HPV-associated subtype showed a better prognosis compared to the other two HPV-negative subtypes (log rank, *p* = 0.0092 and 0.0001; hazard ratio, 1.36 and 1.39). Additionally, within the HPV-negative group, the smoking-related subgroup exhibited worse prognosis compared to the subgroup with high expression in immune response (log rank, *p* = 0.039; hazard ratio, 1.53). The HNSC88 not only enables the identification of HPV-associated subtypes, but also proposes two potential HPV-negative subtypes with distinct prognoses and molecular signatures. This study provides valuable strategies for summarizing the roles and influences of genes in tumorigenesis for identifying molecular signatures and subtypes of HNSC.

## 1. Introduction

Head and neck squamous cell carcinoma (HNSC) is a type of cancer with a high mortality rate worldwide, accounting for over 800,000 new cases and 400,000 deaths annually [1]. The prevalent risk factors contributing to the increased incidence of HNSC include smoking, drinking, human papilloma virus (HPV) infection, and chewing of areca. Among these risk factors, HPV status is a critical prognostic indicator for HNSC. HPV-positive HNSC patients often exhibit more favorable clinical outcomes compared to HPV-negative patients. [2,3,4]. The survival rates of HNSC patients have not significantly improved in recent decades due to the heterogeneity of the disease in terms of etiologies, tumor sites, and genetic characterization. This heterogeneity has posed challenges in the discovery of effective diagnostic, prognostic, and therapeutic biomarkers for HNSC [2,3,4,5]. Due to the heterogeneity of HNSC, some studies have revealed that approximately 60% of HNSC patients with overexpression of epidermal growth factor receptor (EGFR) experienced worse clinical outcomes, but the remaining 40% of patients did not [2,6,7]. Currently, HNSC patients who have failed standard treatments, such as surgery, radiation, and chemotherapy, are lacking alternative treatment options.

An ideal biomarker for cancer includes several characteristics. Firstly, it should exhibit significant expression during the disease state. Secondly, the candidate genes can reveal the underlying mechanisms of the disease. Thirdly, it should be associated with disease prognosis. Additionally, the detection should demonstrate consistency and reliability [8]. Previous research has proposed the benefits of biomarkers and subtypes for cancer diagnosis and prognosis. For example, PSA (prostate-specific antigen) is a well-known biomarker for prostate cancer, used for early detection and monitoring prognosis [9]. In breast cancer, biomarkers such as ER (estrogen receptor), PR (progesterone receptor), and HER2 (human epidermal growth factor receptor 2) are utilized for the identification of intrinsic subtypes [10,11,12,13]. Additionally, the breast cancer 70-gene MammaPrint microarray assay is used to predict patient prognosis [14]. These biomarkers are valuable to decide treatment and predict patient clinical outcomes in breast cancer. However, beyond the distinction of the subgroups of HPV-positive and HPV-negative, there are no established molecular subtypes for HNSC clinical application [2,3,15].

In this study, we aim to develop a strategy for the identification of HNSC molecular signatures and subtypes (Figure 1). We first utilized gene expression between tumor and normal tissues to determine 289 differentially up-regulated DEGs of three microarrays. Then, the two scoring methods, representative score (RS) and perturbative score (PS), were developed to group and select 88 genes (HNSC88) by estimating their relative relations of cellular functions and dysregulation in the tumor state. Finally, the hierarchical clustering was performed to discriminate three HNSC subtypes, and Kaplan–Meier analysis was used to identify their prognostic significances.

## 2. Results

### 2.1. Identification of Molecular Signature for HNSC Using CS, RS, and PS Scoring

To obtain the molecular signature that reveals cellular functions and subtypes of HNSC, three scoring methods were developed to identify candidate genes; namely, Cluster Score (CS), Representative Score (RS), and Perturbation Score (PS). First, we identified 478 DEGs that significantly changed across three microarrays. Based on the general characteristics of biomarker used in clinical medicine [8], only 289 up-regulated DEGs were further analyzed in this study. The CS value (Equation (1)) was calculated to estimate the relative relation of cellular functions between any two genes among the 289 up-regulated DEGs (Figure 2). Next, the 289 DEGs were clustered into 27 groups by using CS values and unsupervised hierarchical clustering. To identify the cellular functions and pathways of the 27 cluster groups in HNSC, we implemented functional enrichment analysis of GOBP, KEGG pathways, and ten hallmarks of cancer, with the hypergeometric test. The results of the enrichment analysis were ranked by enrichment *p*-value, with the top three being presented in Table 1.

Of the 27 cluster groups, only 11 groups (G1 to G11, Figure 2, left dendrogram) contained at least one GOBP with an enrichment *p*-value < 0.05 and had ≥3 gene members involved in that GOBP. The cellular functions of these 11 groups could be summarized into five cancer-related functions for HNSC: tumor immune response (G1 to G3), tumor survival (G4), tumor metastasis (G5 to G7, G10), tumor growth (G8, G9), and tumor metabolism (G11).

For these 11 groups, the RS scoring (Equation (2)) was used to calculate the sum of CS values among the gene members, estimating the genes that were representative of cellular functions within a cluster group. The PS scoring (Equation (3)) was applied to each gene to estimate the influence when the gene was dysregulated in HNSC. Finally, the 88 genes (HNSC88) were selected by ranking the geometric means of RS and PS (Equation (4)). In this study, the HNSC88 gene set was considered the molecular signature for classifying and identifying HNSC tumors. The list of 88 genes (HNSC88) and their corresponding fold change between tumor subgroups and normal tissues are presented in Appendix A.

### 2.2. Unsupervised Clustering of HNSC Subtypes Based on the Expression of HNSC88

To identify HNSC molecular subtypes, we employed HNSC88 and unsupervised hierarchical clustering on 345 HNSC tumors, including full HPV status records (Figure 3). The HPV status was collected by combining the TCGA clinical data with the results of Tang, K. W. et al. [16]. Based on the clustering dendrogram (Spearman’s *ρ* ≥ 0.7) and the gene expression profile of 345 tumors and HNSC88, three primary subtypes were determined (1: purple, 2: pink, and 3: blue). We observed differential distributions of HPV status, smoking history, and TP53 somatic mutations among the three subtypes, which may reflect in the survival rate.

For example, cluster 1 was considered an HPV-negative smoking-related subtype, which included the largest percentage of smoking patients (43%) and showed low expression in tumor immune response (indicated by the yellow frame a). Some studies have indicated that smoking may suppress human immune function, thus facilitating immune evasion by cancer cells and reducing the effectiveness of HNSC treatments [17,18,19,20,21,22]. Cluster 2 was considered an HPV-negative immune-expressed subtype, displaying the highest gene expression in the gene set related to tumor immune response (indicated by the blue frame 1). On the other hand, cluster 3 was classified as an HPV-positive-related subtype, showing the largest percentage of HPV-infected patients compared to cluster 1 and cluster 2, respectively (67% versus 5.7% and 2.2%). The HPV-positive-related subtype also exhibited higher expression in tumor growth, and some studies have indicated that HPV virus may dysregulate the cell cycle to promote cancer cell growth (e.g., CDK4 and MCM2, both members of HNSC88) [2,23]. Furthermore, there was no significant difference between cluster 1 and cluster 2 in the ratio of TP53 somatic mutations (odds ratio was 0.97). However, the frequency of TP53 somatic mutations in both HPV-negative subgroups was higher than that in the HPV-positive-related subgroup (odds ratios were 2.18 and 2.25, respectively). We also observed that the genes and their gene expression involved in tumor metastasis of HPV-negative patients in both subgroups are often up-regulated DEGs or show higher expression, compared to HPV-positive-related patients. This may be one of the reasons why HPV-positive patients have better clinical outcomes (the yellow frame b).

### 2.3. Prognostic Significance of Three HNSC Subtypes

The genome characterization and clinical features mentioned above are reflected in the survival rates of subtypes (Figure 4). Firstly, the cluster 3 subtype (HPV-positive-related) significantly contains more HPV-infected patients who are often TP53 wild-type and non-smokers, showing more favorable survival outcomes. In addition, compared to patients in clusters 1 and 2, HPV-negative patients in cluster 3 present a similar gene expression profile to HPV-positive patients (lower expression in tumor metastasis), which may contribute to a better survival rate (Figure 4, blue line).

We further identified two interesting HPV-negative subtypes (cluster 1 and 2, purple and pink lines) in which the tumor metastasis gene set was significantly up-regulated, leading to adverse prognosis compared to the HPV-positive subtype (log-rank *p*-value = 0.0001 and 0.009; hazard ratio = 1.39 and 1.36, respectively). Moreover, the survival of the HPV-negative smoking-related subtype (cluster 1) was significantly lower compared to cluster 2, further increasing the risk of death (log-rank *p*-value = 0.039; HR = 1.53). Based on the clustering and survival analysis, we identified three subtypes with statistically significant differences in overall survival. One subtype was HPV-positive, associated with favorable survival, while the other two subtypes were HPV-negative, exhibiting poor survival and could be distinguished based on gene expression and smoking behavior.

### 2.4. Immunohistochemistry (IHC) Stain of Clinical HNSC Patients

Based on our domain knowledge, we selected three genes, MMP9, NCF2, and IFI30, to verify their protein expression in tumors and normal tissues using IHC staining (Figure 5). The primary antibodies were purchased from GeneTex (Irvine, CA, USA), with catalog numbers GTX62122, GTX62954, and GTX103967, respectively. The gene MMP9 was used as a positive control and is a well-known biomarker for HNSC [24,25], which showed strong protein expression only in tumor regions (Figure 5A). The other two genes, NCF2 and IFI30, play roles in immune response, antigen presentation, inflammation, cell invasion, and cell survival, and they have been considered as diagnostic and prognostic biomarkers in several cancers [26,27,28,29,30,31]. However, there are few studies that have provided evidence for their roles in HNSC (Figure 5B,C). According to the literature, the gene IFI30 is a novel biomarker for HNSC, and we may have conducted the first study to verify its protein expression in HNSC slides by IHC staining. The protein expression in tumor and normal regions were presented by − (negative) and + (positive) (Figure 5D).

## 3. Methods

### 3.1. Datasets

In this study, we collected 235 HNSC tumor samples and 71 corresponding normal tissues from three GEO microarrays (GSE30784, GSE6791, and GSE9844) for the identification of HNSC signatures [32,33,34]. The platform of the three microarrays is the Affymetrix Human Genome U133 Plus 2.0 Array, which includes 54,675 probes. All raw data (CEL files) were processed using RMA normalization and log2-transformation with the R package “affy” [35]. Next, we mapped the 54,675 probes to 19,213 genes (UniProt, Cambridge, UK) using Affymetrix annotations (version 34). For each gene, we obtained a corresponding expression value from the probes. If a gene was mapped by multiple probes, its expression value was calculated as the average of these probes.

We further collected 520 HNSC tumor samples and 44 normal tissues (level 3 RNA-Seq data with tumor type: 01) from TCGA to validate subtypes [36]. For TCGA samples, we utilized the clinical data, such as HPV status, smoking history, and clinical follow-up information, to annotate HNSC subtypes and conduct survival analysis. Because HPV status is a very important risk factor for HNSC in clinical outcomes, we only selected 345 TCGA patients annotated with HPV status for further subtype analysis. The HPV statuses of 345 patients were obtained and integrated from TCGA clinical data and Tang, K. W. et al. [16]. To evaluate the perturbations of DEGs for signaling pathways, we collected human protein–protein interactions (PPIs) from five databases (BioGRID, Samuel Lunenfeld Research Institute, Toronto, ON, Canada; DIP, University of California, Los Angeles, CA, USA; IntAct, European Molecular Biology Laboratory, Cambridge, UK; MINT, University of Rome Tor Vergata, Rome, Italia; and MIPS, German Research Center for Environmental Health, Neuherberg, Germany) [37,38,39,40,41], and utilized our previous approaches to obtain 267,326 PPIs of 16,596 human proteins [42,43].

### 3.2. Identification of Differentially Expressed Genes (DEGs)

We used the fold change and *p*-value (R package limma) to identify the differentially expressed genes (DEGs) between normal tissues and tumors in the three microarrays. By using the |fold change| ≥ 1.5 and *p*-value < 0.05, we found 2521, 1814, and 847 DEGs in the GSE30784, GSE6791, and GSE9844 datasets, respectively. Based on the previously mentioned characteristics of an ideal biological marker, we only used 289 DEGs that were significantly up-regulated in all three microarrays for further analysis.

### 3.3. Functional Enrichment Analysis

The terms of gene ontology biological process (GOBP) and cellular component (GOCC) were obtained from the Gene Ontology database (http://geneontology.org/). For these GO terms, we employed five relationships to construct the GO trees: “is_a”, “part_of”, “regulates”, “negatively_regulates”, and “positively_regulates”. These GO data were utilized for developing the Cluster Score (CS) and conducting enrichment analysis for gene cluster groups [44,45]. We additionally collected gene sets of 10 cancer hallmarks to analyze the connections between tumor behaviors and 289 DEGs. The cancer hallmarks data were obtained from the Atlas of Cancer Signalling Network (ACSN) database (https://acsn.curie.fr/ACSN2/HMC.html, accessed on 19 July 2023), which offered 93 cancer-related functions and 2652 cancer-related genes [46]. The 330 pathways were also downloaded from the KEGG database and utilized to annotate the potential signaling pathways in which the DEGs are involved. We downloaded the XML pages (KGML) for each KEGG pathway and extracted the gene members of these pathways [47]. The summary of the enrichment analysis is displayed in Table 1.

### 3.4. The Scoring Methods for Identification of HNSC-Related Genes

We have developed the Cluster Score (CS) to estimate the similarity of cellular functions between any two genes out of the 289 DEGs in HNSC tumors. The CS value, along with hierarchical clustering, was employed to group the 289 DEGs based on the relative relations of their cellular functions. The calculation of CS is as follows:(1)$${CS}_{i,j}=\frac{{RSS}_{{BP}_{i,j}}+{RSS}_{{CC}_{i,j}}}{2}+\frac{\sum_{n=1}^{N}{PCC}_{i,j}^{n}}{N}$$
where *RSS_BP_* and *RSS_CC_* represent the relative specificity similarity (RSS) of gene ontology biological process (GOBP) and cellular component (GOCC) between gene *i* and gene *j*, respectively (RSS value ranges from 0 to 1) [48]. *PCC*_*i*,*j*_ denotes the Pearson’s correlation coefficient of gene expression between gene *i* and gene *j*. *N* refers to the number of microarray datasets used (in this case, *N* = 3).

After clustering the 289 DEGs using CS, we developed two scoring methods; namely, the Representative Score (RS) and Perturbation Score (PS), to identify the molecular signatures for the classification of HNSC subtypes. For each gene, the RS value is calculated as the sum of the CS values within each cluster group. This allows us to identify genes that exhibit a high association with every other gene member in terms of cellular functions. On the other hand, to assess the influence of genes (i.e., DEGs) on signaling pathways when dysfunctions occur in HNSC, we utilized the human PPI network to compute the Perturbation Score (PS). The RS and PS are defined as follows:(2)$${RS}_{i}=\frac{\sum_{j=1}^{M}{CS}_{i,j}}{2\times M}$$
where *M* is the number of gene members in the cluster group, and *CS*_*i*,*j*_ is the cluster score between gene *i* and gene *j* in the same cluster group.
(3)$$PS={NFC}_{i}+{NP}_{i}+{NIG}_{i}+{NCG}_{i}$$
where *NFC_i_* is the fold change of gene *i* between normal tissues and tumors; *NP_i_* is the t-statistics *p*-value of gene *i* between normal tissues and tumors; *NIG_i_* represents the number of genes that both interact with gene *i* (i.e., PPIs of gene *i*) and are co-expressed (|Pearson’s *r*| ≥ 0.5) with gene *i*. For example, the protein NCF2 (UniProt: P19878) has 37 PPIs and is co-expressed with 1 gene among them. Therefore, its *NIG_i_* value in one of the datasets is 1. In addition, we also provided *NCG_i_* values for the genes that lacked PPI data. *NCG_i_* is the number of genes co-expressed (|Pearson’s *r*| ≥ 0.7) with gene *i* in the context of HNSC. The PS values of the three microarrays were ranked, averaged, and then normalized to a range of 0 to 1. Then, the geometric mean of the RS value and PS value was employed to select the candidate genes.

The number of selected candidates (*SC*) from each cluster group is provided as follows:(4)$$SC=\frac{G-min(G)}{min(G)}+min(G)$$
where *G* is the number of genes in each cluster group (i.e., G1 to G11). The *min(G)* is the minimum number of members among all cluster groups (here it is 5). Finally, a total of 88 genes (called HNSC88) from 11 groups were selected for classifying HNSC subtypes.

### 3.5. Survival Analysis

We utilized TCGA clinical data, including clinical follow-up, smoking history, and HPV status, to investigate the prognostic significance of HNSC subtypes. The 5-year overall survival (OS) between HNSC subtypes was compared using Kaplan–Meier analysis (R-2.15.3 package survplot) [49]. The log-rank test and Cox proportional hazards regression model (R package survival) were employed to estimate their prognostic differences.

## 4. Discussion

HNSC is a highly heterogeneous cancer, and there are only a few alternative treatment options available for patients who do not respond to surgery or radiotherapy/chemotherapy. Therefore, our aim is to discover molecular signatures that can determine HNSC subtypes with prognostic significance. In this study, we have employed several strategies and made the following findings: (1) We developed two scoring methods to select 88 genes (HNSC88) that were significantly up-regulated in three microarrays; namely, representative score (RS) and perturbative score (PS). (2) HNSC88 is involved in five major cellular functions, including tumor immune response, tumor metastasis, tumor survival, tumor growth, and tumor metabolism, which contribute to the determination of HNSC subtypes. (3) Three HNSC subtypes were identified and showed different prognostic significance of the one HPV-positive and two HPV-negative subgroups. The most distinct difference in molecular signatures between the two HPV-negative subgroups was related to smoking behavior and tumor immune response. Numerous studies have indicated that smoking can suppress the immune system and hinder the effectiveness of HNSC treatment [17,20,50]. We believe that our results not only provide additional evidence for the relationships between HPV status, smoking, and clinical outcomes, but also highlight the detrimental effects of tobacco consumption on HNSC.

However, some limitations and findings still need to be improved and verified. First, our RS and PS scoring rely on the annotations of cellular functions (i.e., GOBP and GOCC) and protein-interaction records to estimate the relationship between two genes. With more comprehensive annotations, we could potentially cluster genes more accurately and discover additional biological functions by considering surrounding gene members. Second, the percentage of patients with TP53 mutation was not significantly different between the two HPV-negative subtypes (cluster 1 and 2). Therefore, we can only suggest the associations among smoking, immune response, and clinical outcomes without providing specific genomic pattern results (e.g., mutation data).

## 5. Conclusions

In summary, we have presented a strategy to select the HNSC molecular signature for predicting three subtypes with prognostic significance. The three subtypes include one HPV-positive subtype with a favorable prognosis, which is consistent with current domain knowledge. The other two HPV-negative subtypes showed differences in smoking behavior and the gene expression related to tumor immune response, and they also exhibited distinct survival rates. Additionally, we identified a novel gene, IFI30, that may be associated with favorable prognosis. Compared to the subtype with the worst survival rate, IFI30 was overexpressed by approximately 2-fold in the subtypes with the best and second-best favorable survival. We believe that our scoring methods and HNSC88 provide an opportunity to develop diagnostic and prognostic markers for HNSC in the future.

## Figures and Tables

**Figure 1 ijms-24-13068-f001:**
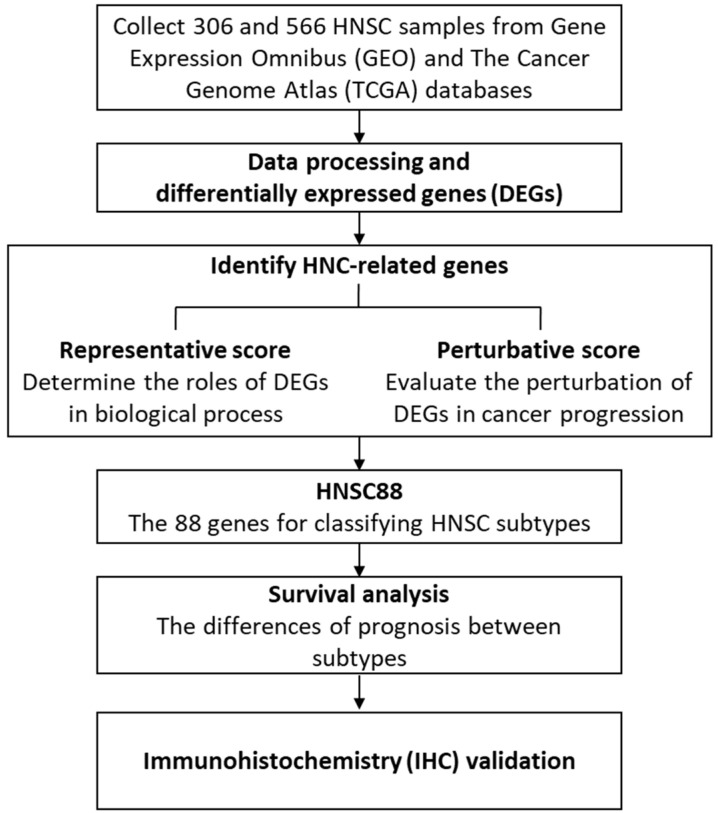
Schematic of identifying HNSC biomarkers.

**Figure 2 ijms-24-13068-f002:**
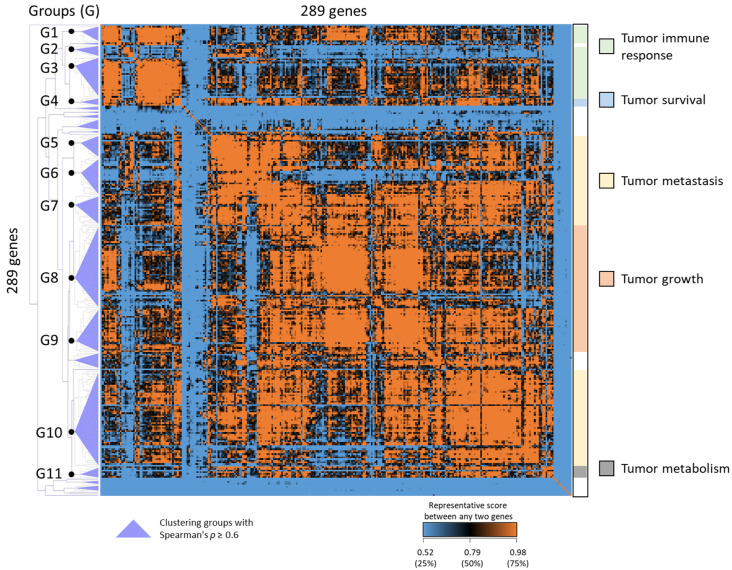
Hierarchical clustering of CS values for 289 significantly up-regulated DEGs. The clustering demonstrates the relative relation of cellular functions between any two genes. The 289 DEGs were classified into 27 cluster groups using a threshold of Spearman’s *ρ* > 0.6 (indicated by the left purple triangle). GOBP enrichment analysis was conducted, and only 11 groups (G1 to G11) were identified, which had at least one significantly enriched GOBP with a *p*-value < 0.05 and ≥3 members. The five cancer hallmarks represented by the 11 groups are displayed in the right color bar.

**Figure 3 ijms-24-13068-f003:**
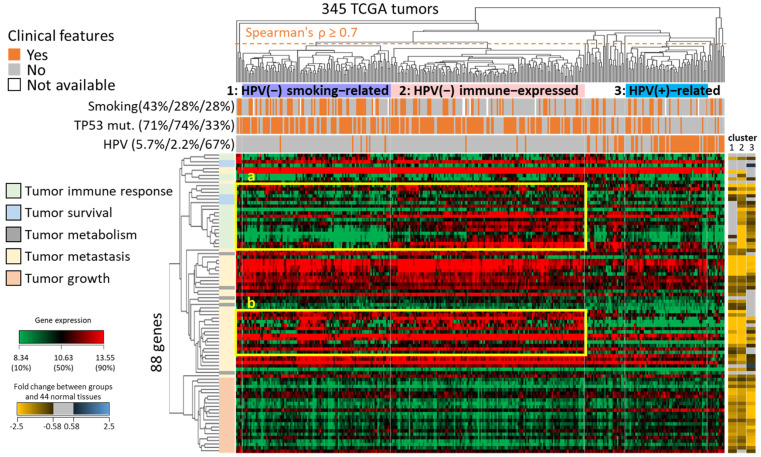
Hierarchical clustering of HNSC88 gene expression for the classification of HNSC subtypes. Unsupervised clustering analysis of gene expression for 88 genes (HNSC88) in 345 HNSC tumors revealed three distinct subgroups (cluster 1, 2, and 3) using Spearman’s *ρ* ≥ 0.7 as the threshold. Above the heatmap, the distribution of three clinical features (i.e., current smoking, TP53 somatic mutation, and HPV status) is displayed with orange color for the individual tumor samples. To the left of the heatmap, the functions of the 88 genes were identified through enrichment analysis (please see Figure 2 and Table 1). The expression fold changes of the 88 genes between the three subtypes and normal tissues are shown on the right.

**Figure 4 ijms-24-13068-f004:**
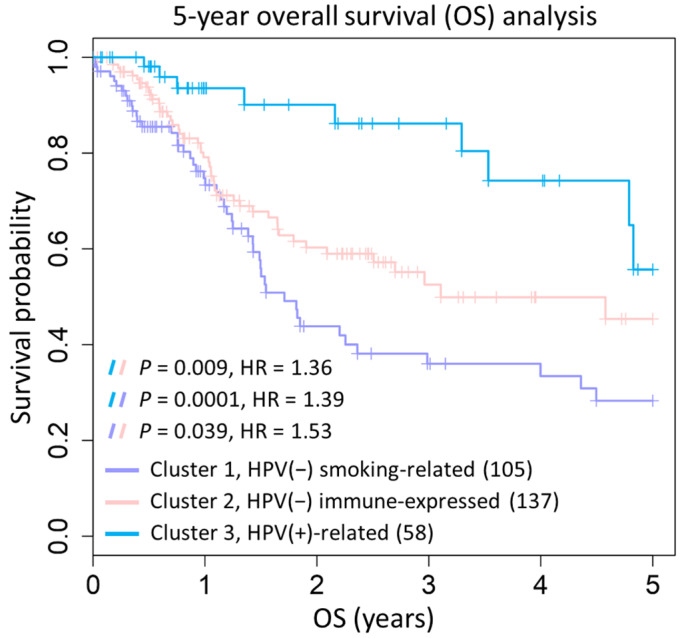
Kaplan–Meier plots of overall survival (OS) for three subtypes identified by HNSC88 in 345 HNSC tumors (blue: HPV-positive; pink: HPV-negative immune-expressed; purple: HPV-negative smoking-related). The *p*-value of log-rank test and hazard ratio between subtypes were performed and displayed.

**Figure 5 ijms-24-13068-f005:**
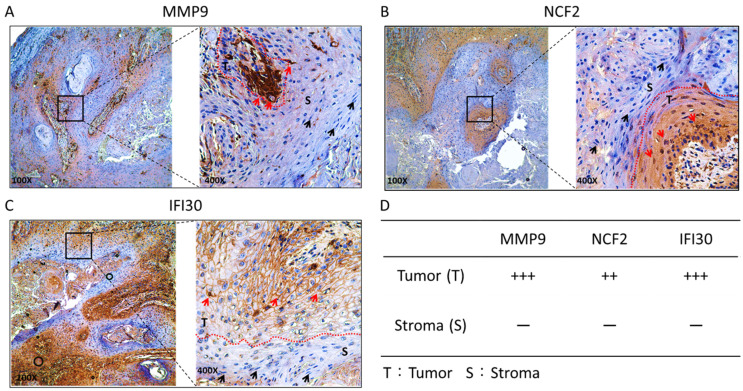
Evaluation of protein expression of (**A**) MMP9, (**B**) NCF2, and (**C**) IFI30 by immunohistochemistry (IHC) staining. The tumor tissue slides were stained with primary antibodies against MMP9, NCF2, and IFI30. Their protein expressions in the cancerous region are indicated by red arrows, whereas those in the normal adjacent region are indicated with black arrows. The red dashed line was used to highlight tumor tissues (i.e., the brown part). (**D**) The intensity of protein expression is presented as − (negative) and + (positive), respectively.

**Table 1 ijms-24-13068-t001:** GOBP, KEGG and cancer hallmark enrichment analysis.

Cluster Group ^#^	Members	GOBP	KEGG Pathways	Summary of Pathways	Cancer-Related Processes	The Hallmarks of Cancer (Summary of Processes)
G1	APOBEC3A, CMPK2, DTX3L, HERC5, IFI30, STAT1	- Defense response to virus; - Response to virus; - Response to other organism;	-	-	-	-
G2	CXCL1, IL1F9, IL8, MMP12, MMP9	- Response to wounding; - Inflammatory response; - Locomotion;	- IL-17 signaling pathway; - Cytokine-cytokine receptor interaction; - Rheumatoid arthritis;	- Signaling molecules and interaction; - Immune disease;	- Cytokines chemokines production; - Tumor growth;	- Activating Invasion and Metastasis; - Tumor-Promoting Inflammation;
G3	DDX60, GBP1, IFI35, IFI6, IFIT1, IFIT3, ISG15, RSAD2, UBE2L6	- Response to virus; - Defense response to virus; - Innate immune response;	- Cytosolic DNA-sensing pathway; - Influenza A; - RIG-I-like receptor signaling pathway;	- Immune system; - Infectious disease: viral;	-	-
G4	CTSL1, NCF2, RGS2, SOD2, TLR2	- Response to lipopolysaccharide; - Response to molecule of bacterial origin; - Cell-type specific apoptotic process;	- Phagosome;	- Transport and catabolism;	-	-
G5	COL4A1, COL4A2, COL5A1, COL5A2, CTHRC1, PXDN	- Collagen catabolic process; - Multicellular organismal catabolic process; - Extracellular matrix disassembly;	- Protein digestion and absorption; - ECM-receptor interaction; - Focal adhesion;	- Digestive system; - Signaling molecules and interaction; - Cellular community—eukaryotes;	- Cell matrix adhesions; - ECM; - EMT regulators;	- Activating Invasion and Metastasis;
G6	INHBA, LAMC2, MMP1, MMP10, MMP3, PLAU, PTHLH, TGFBI	- Extracellular matrix organization; - Extracellular structure organization; - Cellular component movement;	- TGF-beta signaling pathway; - IL-17 signaling pathway; - Prostate cancer;	- Signal transduction; - Immune system; - Cancer: specific types;	- ECM; - Tumor growth; - Matrix regulation;	- Activating Invasion and Metastasis; - Tumor-Promoting Inflammation;
G7	ACVR1, CLIC4, DSG2, FAT1, FNDC3B, ITGAV, SNAI2	- Vasculature development; - Cardiovascular system development; - Circulatory system development;	-	-	- EMT regulator;	- Activating Invasion and Metastasis;
G8	CDC45, CDC6, CDK4, CHEK1, CKS1B, DTL, FEN1, FOXM1, GINS1, MCM2, RFC4, TPX2, UHRF1	- Cell cycle process; - Cell cycle; - Interphase;	- DNA replication; - Cell cycle; - p53 signaling pathway;	- Replication and repair; - Cell growth and death;	- S-CC phase;	- Evading Growth Suppressors;
G9	AURKA, BIRC5, CDC20, CEP55, ECT2, KIF2C, KIF4A, NUP155, TRIP13	- Nuclear division; - Mitosis; - Organelle fission;	-	-	-	-
G10	CDH3, CEBPB, DFNA5, GJA1, HOMER3, JUP, KLF10, KLF7, KRT17, MSN, MYO1B, PANX1, PRNP, RRAS2, SLC16A1	- System development; - Locomotion; - Response to endogenous stimulus;	- Regulation of actin cytoskeleton; - Arrhythmogenic right ventricular cardiomyopathy (ARVC); - Autophagy—animal;	- Cell motility; - Cardiovascular disease; - Transport and catabolism;	- EMT regulator; - Cell-cell adhesions;	- Activating Invasion and Metastasis;
G11	ACOT9, FKBP9, GALNT2, UBE2Q2, VKORC1	- Peptidyl-amino acid modification;	-	-	-	-

*^#^* The column colors of the Cluster Group correspond to the five cancer hallmarks in Figure 2.

## Data Availability

The three HNSC microarrays (GSE30784, GSE6791 and GSE9844) were collected from Gene Expression Omnibus (GEO). The RNA-Seq of HNSC tissues and clinical information were collected from TCGA-HNSC cohort of Genomic Data Commons Data Portal database (https://portal.gdc.cancer.gov/). Kaplan-Meier analysis was performed by using R tool which could be obtained from R Project (https://www.r-project.org/). All relevant materials are provided in the manuscript. The list of 88 genes (HNSC88) and their corresponding fold change between tumor subgroups and normal tissues are presented in Appendix A.

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
