# Peer review of "Identification of the HNSC88 Molecular Signature for Predicting Subtypes of Head and Neck Cancer"

_ijms, 2023, doi:10.3390/ijms241713068_

Round 1

Reviewer 1 Report

In this work Chuang et al. provides a strategy for identifying molecular signatures and subtypes of HNSC. Although the considerable done amount of work, I would like to point out some minor points that need to be considered and integrated (or at least justified) for the purpose of publishing the presented manuscript:

- line 36: please use the past tense

- line 60: please define the EGFR term

- line 99: remove "the"

- lines 107-108: how do you behave if PPI data are missing?

- line 113: how did you choose the threshold?

- section 2.3: Being this part in the methods section, could you give more details about how did you collected data from the cited databases?

- line 127: why you did not consider the third gene ontology class, i.e. the molecular function, in the formulation of CS score?

- line 137, 141, 153: members instead of member

- line 140: please correct "are defined" instead of "is defined"

- lines 144-145: could you better clarify the definition of NIG?

- line 154: please remove "and" from the beginning of the sentence.

- please correct throughout all the manuscript all the "HNC" terms that should be "HNSC". 

- section 3.2: it's not clear which are the frame 1 and 2 in figure 3

According to the comments and suggestions, some minor is just suggested to improve the quality of English language in the manuscript.

Author Response

We have carefully revised the manuscript according to all the suggestions from the reviewers. Our point-to-point responses are included as below. Please note that all changes in the revised manuscript are marked red to facilitate the review. The revised manuscript has been uploaded as an attachment.

Response to Reviewer 1 Comments

In this work Chuang et al. provides a strategy for identifying molecular signatures and subtypes of HNSC. Although the considerable done amount of work, I would like to point out some minor points that need to be considered and integrated (or at least justified) for the purpose of publishing the presented manuscript:

Response: Thank you very much for your constructive comments on our work.

Point (1): line 36: please use the past tense

Response (1): Thank you very much for editing of English language. We corrected the mistake of English language. (lines 40)

Point (2): line 60: please define the EGFR term

Response (2): Thank you for the advice. In this revised manuscript, we added the definition of EGFR term. (lines 63)

Point (3): line 99: remove "the"

Response (3): Corrected. (lines 102)

Point (4): lines 107-108: how do you behave if PPI data are missing?

Response (4): We used co-expression analysis (NCG values) of microarray data to assess the correlation between two genes when PPI data was unavailable. The co-expression was determined using the Pearson correlation coefficient (Pearson's r) and the gene expression of the two genes. The values of Pearson's r range from -1 to 1. Positive co-expression (Pearson's r >0) refers to the gene expression trends of two genes are both up-regulated or down-regulated across all samples. Conversely, negative co-expression indicates that the gene expression trends of the two genes are opposite (one up and one down). In this study, we utilized the Perturbation Score (PS) to evaluate the impacts of genes in the HNSC state through PPI data and co-expression analysis. (lines 156 to 160)

Point (5): line 113: how did you choose the threshold?

Response (5): In this study, we applied a criterion of |fold change| ≥1.5 and p-value <0.05, because the smallest threshold for fold change among the three datasets we utilized is 1.5 (i.e., GSE307841). There were also some other studies used |fold change| ≥1.5 and p-value <0.05 as the threshold to identify differentially expressed genes2-6.

Point (6): section 2.3: Being this part in the methods section, could you give more details about how did you collected data from the cited databases?

Response (6): Thank you very much for the advice and suggestions. We have revised the statements in Section 2.3 to provide more details about the process of collecting data from the referenced databases. (lines 121 to 133)

Point (7): line 127: why you did not consider the third gene ontology class, i.e. the molecular function, in the formulation of CS score?

Response (7): In this study, our focus was on identifying a gene set (i.e., 88 genes; HNSC88), to investigate the mechanisms and subtypes of HNSC. Therefore, we prefer to predict and interpret tumor cell behaviors based on a set of genes (such as pathways or biological processes) rather than individual gene-gene/protein-protein interactions. The gene ontology molecular function (GOMF) annotates the specific molecular functions of a gene, such as chemical reactions or molecular binding mechanisms. However, it may not be suitable for recognizing the outcomes of perturbations caused by a set of genes in complex disease.

Point (8): line 137, 141, 153: members instead of member

Response (8): Corrected. (lines 149, 153 and 168)

Point (9): line 140: please correct "are defined" instead of "is defined"

Response (9): Corrected. (lines 152)

Point (10): lines 144-145: could you better clarify the definition of NIG?

Response (10): Thank you very much for your suggestions. In this revised manuscript, we added new statements to clarify the NIG value. (lines 156 to 160)

NIGi represents the number of genes that both interact with gene i (i.e., PPIs of gene i) and are co-expressed (|Pearson's r| ≥0.5) with gene i. For example, the protein NCF2 (UniProt: P19878) has 37 PPIs and is co-expressed with 1 gene among them. Therefore, its NIGi value in one of the datasets is 1. In addition, we also provided NCGi values for the genes that lacked PPI data. NCGi is the number of genes co-expressed (|Pearson's r| ≥0.7) with gene i in the context of HNSC. The NIGi and NCGi values would be calculated in each of the three datasets, respectively. These values would be ranked, averaged, and then normalized to a range of 0 to 1 in order to compose the Perturbation Score (PS).

Point (11): line 154: please remove "and" from the beginning of the sentence.

Response (11): Corrected. We remove "And" from the beginning of the sentence. (lines 199)

Point (12): please correct throughout all the manuscript all the "HNC" terms that should be "HNSC".

Response (12): Corrected. In this revised manuscript, we changed the abbreviation "HNC" to "HNSC". (lines 214, 215, 219, 235, 244, 245, 272, 275, 301, 302 and 364)

Point (13): section 3.2: it's not clear which are the frame 1 and 2 in figure 3

Response (13): Thank you for your valuable suggestions. In this revised manuscript, we edited Figure 3 in section 3.2 by changing the blue frame to a yellow frame in order to emphasize the differences in gene expression. We also revised the statements "(indicated by the blue frame 1)" to "(indicated by the yellow frame a)" and "(the blue frame 2)" to "(the yellow frame b)" to correspond with Figure 3. (lines 225 and line 242; Figure 3)

Reviewer 2 Report

The purpose of this study was to develop a strategy for identification of HNSC molecular signatures and subtypes-

The study covers some issues that have been overlooked in other similar topics. The structure of the manuscript appears adequate and well divided in the sections. Moreover, the study is easy to follow, but some issues should be improved. Some of the comments that would improve the overall quality of the study are:

I-) Authors must pay attention to the technical terms acronyms they used in the text;

II-) Conclusion Section: This paragraph is missing. Please add it.

Author Response

We have carefully revised the manuscript according to all the suggestions from the reviewers. Our point-to-point responses are included as below. Please note that all changes in the revised manuscript are marked red to facilitate the review. The revised manuscript has been uploaded as an attachment.

Response to Reviewer 2 Comments

The purpose of this study was to develop a strategy for identification of HNSC molecular signatures and subtypes. The study covers some issues that have been overlooked in other similar topics. The structure of the manuscript appears adequate and well divided in the sections. Moreover, the study is easy to follow, but some issues should be improved. Some of the comments that would improve the overall quality of the study are:

Response: Thank you very much for acknowledging our works.

Point (1): Authors must pay attention to the technical terms acronyms they used in the text.

Response (1): Thank you very much for the suggestions. In this revised manuscript, we corrected the acronym to ensure consistency, such as changing terms from "HNC" to "HNSC".

Point (2): Conclusion Section: This paragraph is missing. Please add it.

Response (2): Thank you for the advice. In this revised manuscript, we added the Conclusion section in line 323 to 333.
